# Multi-resolution deep learning characterizes tertiary lymphoid structures and their prognostic relevance in solid tumors

Mart van Rijthoven [1✉], Simon Obahor [2], Fabio Pagliarulo[2], Maries van den Broek [2], Peter Schraml[3], Holger Moch [3,4], Jeroen van der Laak [1], Francesco Ciompi[1,6] & Karina Silina [5,6]

## Abstract

**Background** Tertiary lymphoid structures (TLSs) are dense accumulations of lymphocytes in inflamed peripheral tissues, including cancer, and are associated with improved survival and response to immunotherapy in various solid tumors. Histological TLS quantification has been proposed as a novel predictive and prognostic biomarker, but lack of standardized methods of TLS characterization hampers assessment of TLS densities across different patients, diseases, and clinical centers.

**Methods** We introduce an approach based on HookNet-TLS, a multi-resolution deep learning model, for automated and unbiased TLS quantification and identification of germinal centers in routine hematoxylin and eosin stained digital pathology slides. We developed HookNet-TLS using n = 1019 manually annotated TCGA slides from clear cell renal cell carcinoma, muscle-invasive bladder cancer, and lung squamous cell carcinoma.

**Results** Here we show that HookNet-TLS automates TLS quantification across multiple cancer types achieving human-level performance and demonstrates prognostic associations similar to visual assessment.

**Conclusions** HookNet-TLS has the potential to be used as a tool for objective quantification of TLS in routine H&E digital pathology slides. We make HookNet-TLS publicly available to promote its use in research.

## Plain Language Summary

Tertiary lymphoid structures (TLS) are dense accumulations of immune cells within a cancer. They have been associated with patient survival and treatment effectiveness. Quantification of TLS in cancer microscopy images may therefore aid clinical decision-making. However, no consensus for defining TLS in such images exists leading to inconsistent and variable findings across different labs and studies. We developed a computational tool for automated and objective TLS quantification in cancer images. The tool, called HookNet-TLS, integrates information from multiple image resolutions, which resembles the process of how a pathologist would identify these structures using a microscope. HookNet-TLS detected TLS similarly to trained researchers in three different tumor types. We provided access to HookNet-TLS to facilitate its development and use for TLS assessment in clinical decision-making and research into the role of TLS in cancer.

---

[1] Pathology Department, Radboud University Medical Center, Nijmegen, Netherlands. [2] Institute of Experimental Immunology, University of Zurich, Zurich, Switzerland. [3] Department of Pathology and Molecular Pathology, University Hospital Zurich, Zurich, Switzerland. [4] Faculty of Medicine, University of Zurich, Zurich, Switzerland. [5] Institute of Pharmaceutical Sciences, Swiss Federal Institute of Technology, Zurich, Switzerland. [6] These authors contributed equally: Francesco Ciompi, Karina Silina. ✉email: mart.vanrijthoven@radboudumc.nl

Tertiary lymphoid structures (TLSs) are ectopic lymphoid organs that develop in peripheral tissues in response to chronic inflammation. TLSs resemble follicles of secondary lymphoid organs, such as lymph nodes, both in structure and function including priming of antigen-specific lymphocytes[1]. The development of TLSs has been described in several solid tumors in association with improved survival and response to immunotherapy, especially when TLSs are mature, i.e., harboring a germinal center (GC)[1–3]. Thus, TLSs have been proposed to be in situ hubs of anti-tumor immunity, and their quantification holds promise as a novel biomarker for patient risk and treatment stratification[1].

Three main approaches are used to quantify TLS presence in the tumor microenvironment, namely transcriptional profiling, flow cytometry and histology-based assessment. Quantification of TLS-associated transcripts may be used as surrogates for TLS quantification in patients for which histological samples are not available. However, considerable differences in gene signatures associated with TLSs in different organs have been reported[4–6]. Furthermore, the often-used TLS genes are not TLS-specific. For example, CXCL13, a key chemokine involved in TLS development, is also expressed by tumor-infiltrating T cells outside TLSs[7] or even tumor cells themselves[8,9]. We recently demonstrated a poor correlation between TLS-associated transcripts from tumor core biopsies and TLS density in corresponding diagnostic slides[10], suggesting that RNA quantification may provide some but not complete information on TLS development in the tumor microenvironment. Alternatively, flow cytometry allows quantification of cell types significantly enriched in TLS, such as T follicular helper cells[11] as well as follicular and germinal center B cells[12], in tumor tissue digests. However, how well these cell types reflect TLS density and maturation stage in different tumors is not known. Furthermore, tumor single cell suspensions are not broadly available restricting the use of flow cytometry for TLS detection to a narrow set of studies.

Histopathology diagnostics, primarily based on hematoxylin and eosin (H&E) staining, is routinely used in clinical practice. Morphological analysis in H&E allows for direct quantification of TLSs as dense aggregates of lymphocytes with a distinguishable contour in contrast to unorganized inflammation, as well as the detection of GCs owing to the distinct morphology of centrally located proliferating blasts. Studies reporting histological TLS assessment in various cancer types suggest that different peripheral organs significantly differ in their capacity to support TLS development[3]. However, the objective and reliable comparison of TLS development across different studies is hampered by a broad range of varying histological TLS definitions in H&E staining as well as by immunohistochemistry-based detection of various TLS-associated cell types such as B cells (CD20), dendritic cells (DC-LAMP), high endothelial venules (PNAD), or T cells (CD3, CD8) to quantify TLS[12–15]. Considering that immunostaining is not routinely available, an automated method for TLS quantification in H&E slides can offer cost-effectiveness, objectivity, and reproducibility in TLS research and clinical analysis. In this study, we assessed whether the development of a computer algorithm for automatic TLS quantification in H&E images could satisfy these requirements.

In recent years, the availability of high-throughput scanning technology has enabled the digitization of whole-slide images (WSIs) and the advent of digital pathology. This, in turn, has spurred the development of various artificial intelligence (AI) approaches for WSI tissue analysis. Deep learning has been successfully applied to several fields of medical imaging[16], to diagnostic tasks in pathology[17–20], as well as for the quantification and discovery of biomarkers in immuno-oncology[21].

Owing to their specific morphology, immune infiltrates can be detected by deep learning. Our recently developed AI model called HookNet[22] successfully segmented TLSs and GCs as well as other tissue structures in H&E-stained WSIs of lung cancer. HookNet has been previously extensively compared to U-Net[23], a widely acknowledged segmentation model for medical image analysis, and demonstrated superior performance[22]. The unique feature of HookNet is its integration of multiple image resolutions to produce segmentation output. By incorporating high-resolution details with contextual information from lower resolution, mimicking how pathologists examine histology slides, this approach demonstrated an improvement over single (high)-resolution segmentation tested on a limited set of cancer WSIs[22]. Recent studies have developed alternative AI approaches for TLS quantification based on lymphocyte detection[24,25]. However, such approaches learn TLS recognition by means of individual cell detection and need additional post-processing steps that combines individual cells to produce an object-based structure. Furthermore, these methods are therefore also not able to explicitly consider the detection of GCs, which are a crucial prognostic and predictive component of TLS[2,3].

In this study, we build upon our previous work to enable robust TLS analysis in basic research and potential clinical application. We report the development of a deep learning-based model for standardized, objective, and automated quantification of TLSs across different H&E-stained tissues and sample sources, named HookNet-TLS. We demonstrate the robustness and generalizability of HookNet-TLS by leveraging large training datasets as well as independent validation and test sets sourced from The Cancer Genome Atlas (TCGA). As an additional verification, we aimed to assess the prognostic correlations of HookNet-TLS predictions by using three tumor cohorts: lung squamous cell carcinoma (LUSC), clear cell renal cell carcinoma (KIRC), and muscle invasive bladder cancer (BLCA), with known differences in their prognostic associations for TLSs[8,10,26]. We generated manual annotations of TLSs and GCs based on visual interpretation of H&E staining and benchmarked such annotations by matched IHC staining in an independent set from the University Hospital Zurich (USZ). We measured the inter-observer variability to investigate the subjectivity in human-based TLS detection and assessed the model's performance in this context. Furthermore, we compared HookNet-TLS with a state-of-the-art AI approach for object detection, named Faster R-CNN[27], to validate the benefits of a multi-resolution segmentation-based approach versus an approach specifically designed to detect objects in computer vision applications. Finally, we investigated the prognostic correlations of HookNet-TLS predictions, which closely mimicked the results from manual TLS quantification. We made our code and algorithm publicly available as a web-based tool to foster further development of the model for potentially pan-cancer applications and beyond.

## Methods

### Generation of ground truth annotations

*TCGA data.* We downloaded 1481 diagnostic WSIs and the corresponding clinical data for three tumor cohorts from the TCGA consisting of 480 LUSC, 405 BLCA, and 514 KIRC patients. All analyzed H&E images were scanned at 40 × magnification in Aperio svs format. An initial manual quality control was performed, and the following image exclusion criteria were applied: poor tissue or image quality (e.g., presence of large air bubbles, excessive pen marks or out-of-focus areas) and the absence of adjacent normal tissue, since TLSs mainly develops in the periphery of these solid tumors[8,10]. As a result, 1019 slides passed the quality control (BLCA n=345, KIRC n=299, LUSC

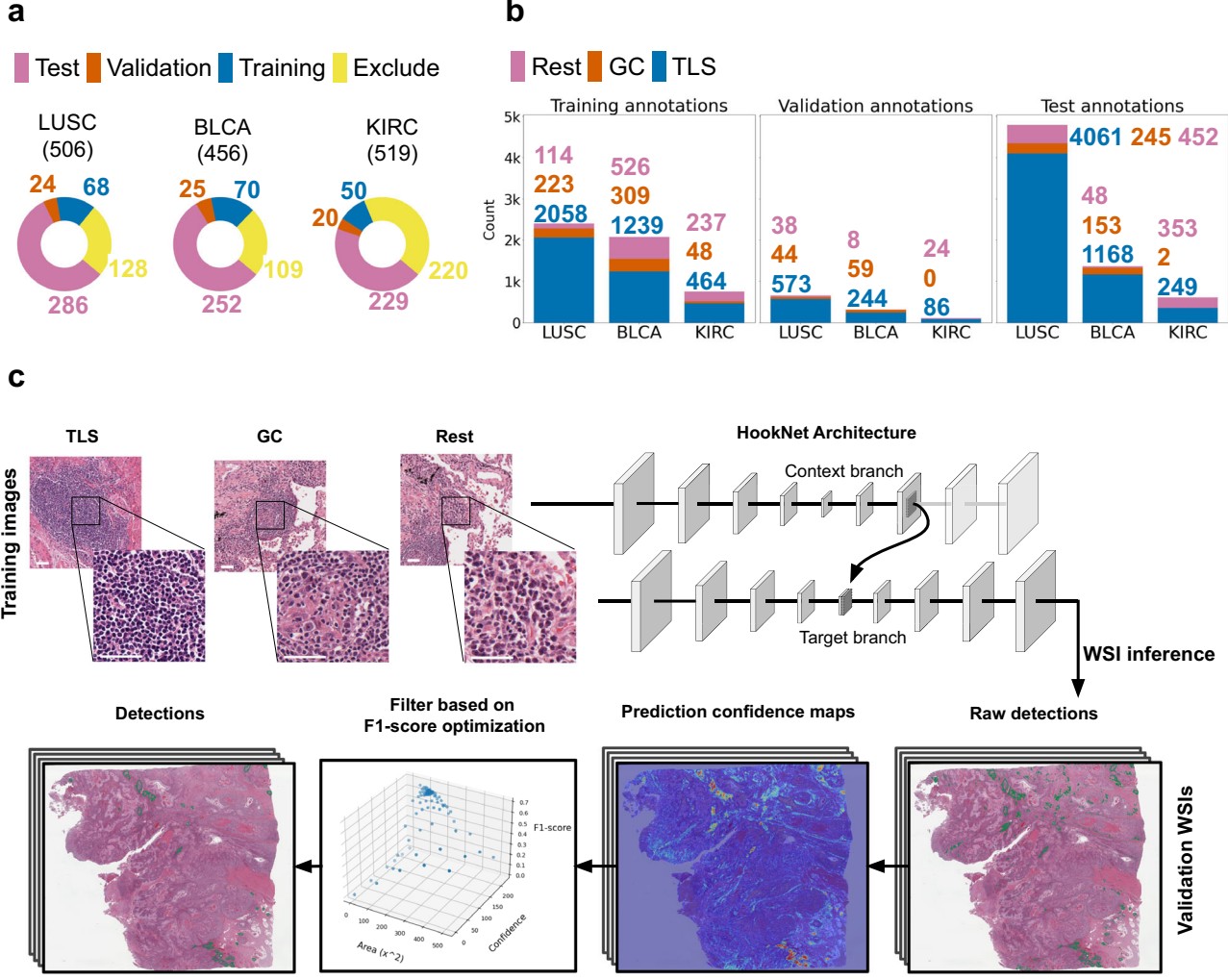

**Fig. 1 Data and HookNet-TLS pipeline. a** All diagnostic slides from the Cancer Genome Atlas (TCGA) muscle invasive bladder cancer (BLCA), clear cell renal cancer (KIRC), and lung squamous cell carcinoma (LUSC), except for those that lacked adjacent normal tissue or had poor scan quality, were split into three independent sets for model training, validation, and testing. **b** The number of annotations per class per dataset. All selected images were exhaustively annotated for tertiary lymphoid structures (TLSs), germinal centers (GCs) and lymph nodes, and sparsely for various other tissues (Rest), which included disorganized infiltrates (inflammation), stroma, tumor, and various parenchymal regions. **c** A schematic representation of the HookNet-TLS model architecture including multi-resolution patch based training using the training dataset, and post-processing steps including whole-slide image inference and object detection optimization using the validation dataset (see Methods section for more details). Scale bars = 50 μm.

n=375) and were exhaustively (i.e., all objects in the whole slide) manually annotated by delineating the border of lymph nodes, TLSs and GCs using the QuPath software (v0.3.2.)[28], and divided into three independent datasets for model training (n=188), validation (n=69), and testing (n=767), respectively (Fig. 1). This allowed for an objective analysis of test cases that were not exposed during training. The division was based on patient ID and the annotated TLS/GC counts per image. The number of slides per set were chosen as such to balance between obtaining a comprehensive set of training annotations while keeping large cohorts independent from the training process in order to test the prognostic relevance of the predicted regions. Finally the purpose of the small validation set was to adjust post-processing parameters of the model, while still keeping it independent from the training process.

Annotated lymph node regions were excluded from all downstream processes due to their high morphological resemblance to TLS. To ensure a balanced distribution of training annotations between different tumor types, we had to include the majority of TLS-high KIRC images in the training set.

Non-exhaustive (i.e., only some objects in the whole slide) annotations were generated for inflammation (unorganized immune cell infiltration), tumor, and other tissue, which included normal parenchyma of the different organs and were collectively labeled as Rest. Different tumor types and slide sets were annotated by, in total, three different annotators trained in the histopathology of TLS to mimic the real-life variability due to subjective TLS quantification in different clinical centers (Supplementary Figure 1).

*USZ data*. We selected 15 tumor samples (7 KIRC, 8 LUSC) with high TLS density from the pathology archive of the Department of Pathology and Molecular Pathology at the University Hospital Zurich (USZ cohort). Patient analyses were conducted according to the Declaration of Helsinki. Ethical approval for performing research on anonymized, archival patient material was obtained from the cantonal ethics commission Zurich (BASEC no. 2022-01854). Patient consent was not obtained, but not necessary due to data anonymization. For each USZ sample, FFPE tumor blocks were used to prepare three 2-μm thick near-serial sections. These

were immunostained for CD20 (clone L26, Leica, CD20-L26-L-CE) and CD23 (clone SP3 Abcam, ab16702) or stained by H&E. All stainings were performed using the Discovery Ultra and BenchMark Ultra automated systems (Ventana Medical Systems) according to the standard operating procedure for diagnostic staining at the USZ. Whole slides were scanned using a Nano-Zoomer 2.0-HT C9600 (Hamamatsu).

In this dataset, we performed two sets of annotations in two independent sessions. First, manual blinded (i.e., without the support of IHC) annotations of H&E slides were generated by following the same procedure as for the TCGA images. Second, the same H&E slides were re-annotated in an IHC-guided manner by using the IHC-stained images as the reference standard. Thus, objective ground truths of TLSs and GCs were generated in H&E, based on the presence of dense aggregates of $CD20^+$ B cells and $CD23^+$ follicular dendritic cell networks within B cell clusters, respectively. All annotations were made using the QuPath software (v0.3.2.)[28] by one trained annotator (Supplementary Figure 1). We used the IHC-guided annotations to evaluate the overall accuracy of manually generated blinded TLS and GC annotations in H&E images. The performance was computed using the F1-score (see Section Model assessment for details).

## Multi-resolution deep learning to characterize TLSs and GCs

*Model development.* The HookNet model[22] is designed to take advantage of both contextual information and high-resolution details to accurately identify structures that may be visible only at lower magnifications, as well as to capture subtle differences at high magnifications. The model is composed of two encoder-decoder U-Net[23] branches, known as the context and target branches. Here we optimized the original model to create HookNet-TLS with improved efficiency, speed and reduced computational cost by reducing the number of parameters from approximately 50 million to approximately 25 million. The reduction was made by reducing the number of filters in each layer of the neural network, which empirically did not have a negative impact on the performance.

In contrast to the approach taken in the original HookNet model[22], our implementation encompasses three major extensions: (1) generation of Whole Slide Image (WSI) segmentation maps, (2) generation of confidence maps, and (3) extraction of detected objects from these maps. Specifically, we employ a sliding window approach, wherein tiles spanning the entire WSI are extracted and subjected to segmentation predictions. Subsequently, these tiles are assembled to construct the complete WSI segmentation map.

For the extraction of TLS and GC detection objects, we utilize the wholeslidedata python package[29] in conjunction with contour finding through the opencv-python python package. Through these extensions, we effectively transform the original HookNet segmentation model into a detection model.

*Training HookNet-TLS and Faster-RCNN.* We used all annotations from the selected TCGA training dataset for training HookNet-TLS to predict the segmentation target classes: TLS, GC and Rest. We followed the training procedure described in[22], which performed best for TLS segmentation and trained HookNet-TLS using concentric patches extracted with 0.5 μm/px and 2.0 μm/px spacing. Both patches had dimensions of 284x284x3 pixels. The model was trained using 'He normal'[30] initialized weights, batch normalization, RELU activations, L2 regularization, and cross-entropy loss. Patch sampling from WSIs was performed using a pixel-count-based strategy, as done in[22], for all classes to ensure that the model has sufficient exposure to

each class. Based on our previous work, wherein we found that utilizing only one loss from the high-resolution branch resulted in the best configuration for TLS and GC segmentation, we trained the model accordingly. For a complete description of HookNet and further training details, please refer to original HookNet model[22].

We used the Detectron2 framework[31] to train a Faster R-CNN with ResNet50 as the encoder. A batch size of 12 was used, and a custom Detectron2 dataloader was implemented using the wholeslidedata package[29]. This data loader allowed for efficient patch sampling from WSIs and enabled us to apply the same data augmentations used in training HookNet. Patches were extracted at 2.0 μm/px with a patch size of 256x256x3 pixels. Furthermore, the default settings from Detectron2 were used to train the model for 20,000 iterations which allowed the model to converge.

*Post-processing.* Thresholding probability scores are standard practice in object detection frameworks. HookNet-TLS generates a prediction probability (confidence) value for each pixel of the WSI instead of a single probability value of an object. Candidate objects were accepted by evaluating an area including pixels with a probability value equal to or greater than a specified threshold. To find the optimal threshold for the detection of true positive prediction objects, we performed a two-dimensional search over the probability and size dimensions using the F1-score as the optimization metric exclusively using the validation dataset (Fig. 1c). To allow us to use integer values in the search, we mapped the probability scores from the floating-point range 0-1 to the integer range 0-255 and considered squared areas up to $255^2$. Faster-RCNN produces a single probability per object. We tuned object thresholds by optimizing the F1 score on the validation set. Standard Faster-RCNN uses non-maximum suppression to reduce overlapping detections, and overlapping objects resulting from the sliding window method, used for full WSI detection, were merged. Finally, because GC objects are predicted independently from TLS objects, we eliminated GC-predicted objects that did not overlap with at least 50% of a TLS.

*Model assessment.* For the purpose of the model evaluation, we used the F1-score:

$$F1 = \frac{2 * precision * recall}{precision + recall} \qquad (1)$$

$$precision = \frac{TP}{TP + FP} \qquad (2)$$

$$recall = \frac{TP}{TP + FN} \qquad (3)$$

The F1 scores were evaluated for every slide in the independent test datasets from the TCGA and USZ. A ground truth annotation was considered a true positive (TP) if it had at least 50% overlap with a prediction. Ground truths not covered by a prediction were counted as false negative (FN). Predictions not covering a ground truth were considered as false positive (FP). Comparisons of F1-scores between groups were done by paired Mann-Whitney *U* test. Correlation between the overall F1-score and the number of training annotations was performed by Spearman correlation analysis.

*Extraction and analysis of encoding features.* The encoding features of each region from manual annotations as well as predictions were extracted from patches centered on each region at the lowest level of the target branch of the HookNet-TLS model. This is the level with the lowest encoding space, including the contextual features. To reduce the feature space, we max-pooled a

10x10x440 feature map from the lowest level of the target branch, resulting in 440 features that contained 85 identical feature values. Therefore in total, 355 distinct features encoded differences across the TLS and GC regions.

We used UMAP to explore the differences between TLS and GC features in a 2-dimensional space and used the Rphenograph algorithm[32] to perform unsupervised clustering of the regions based on the 355 features. With k=50 as the number of nearest neighbors we obtained 14 different clusters among which one–cluster 7, was significantly enriched in the GC class compared to the TLS class in all tumor types. The TLS annotations clustering together with GCs in cluster 7 were retrospectively verified by visual inspection of images to contain a GC.

*Hard-negative mining.* The exhaustive nature of the TLS and GC annotations in our images allowed us to automatically identify and assign false positive predictions from the initial training round as the Rest class to increase the training dataset for subsequent training iterations via the so-called hard-negative mining approach. We attempted two additional training iterations, which included the following steps: (1) Train the model using the initial training set made by the annotators on the TCGA data. (2) Apply the trained model to the full slides in the training set. (3) Evaluate the model against the reference standard made for the TCGA dataset and identify false positives for TLS and GC annotations. (4) Extend the training set with the false positives as Rest annotations and retrain the model again with randomly initialized weights. (5) Evaluate the performance of the retrained model.

**Quantitative histology.** The number of TLS and GC counts and area values were extracted from the manual annotations generated using the QuPath software and from object segmentation maps of HookNet-TLS. The total tissue area was measured by ImageJ for each slide by setting a pixel intensity threshold value common for all slides. Of note, the used intensity threshold excluded large fatty tissue regions surrounding mostly the BLCA specimens. The mean TLS size and TLS density (count per total tissue area) was calculated for each slide. The following density parameters were obtained: the total TLS density and GC density. The histological parameters (TLS and GC size and density) were compared between manual annotations and HookNet-TLS predictions in the independent test set images using paired Mann-Whitney $U$ test.

**Multiplex immunofluorescence.** Near-serial FFPE slides from the USZ KIRC set were processed manually for multiparameter immunofluorescence. Slides were heated at 55 °C for 2 h, then placed in Trilogy Solution™ (CellMarque) in a pressure cooker for 15 min according to the manufacturer's protocol. The slides were cooled for 15 min and washed in running tap water. Slides were treated with 3% $H_2O_2$, washed in 0.1% Triton X-100/PBS and blocked using 2% BSA/0.1% Triton X100/PBS. Antibodies were diluted in 1% BSA/0.1% Triton X-100/PBS. Samples were incubated with primary antibodies overnight at 4 °C or for 3 h at room temperature, washed 3 times and incubated with secondary antibodies for 1 h at room temperature.

For detection, secondary antibodies labeled with horseradish peroxidase (all with reduced cross-reactivity from JacksonImmunoresearch) in combination with 7plex Opal system (Akoya) were used according to the manufacturer's protocol. Slides were incubated in DAPI (0.5 µg/ml) (Life Technologies) and mounted using ProlongDiamond medium (Life Technologies). The following antibodies were used: CD3 (clone SP7, Thermo Scientific,

RM-9107), CD21 (clone 2G9, Leica, CD21-2G9-L-CE), CD23 (clone SP3, Abcam, ab16702), CD20 (clone L26, Leica, CD20-L26-L-CE), DC-LAMP (clone 1010E1.01, Dendritics, DDX0191). Whole slides stained by IF were imaged using multispectral imaging system Vectra 3.0 (PerkinElmer). 20x high power fields were selected to image all dense lymphocytic aggregates multispectrally in each slide and images were processed using the Inform software v2.6 (Akoya).

**Survival analysis.** Clinical data corresponding to the patients included in the histological analysis was obtained from the GDC data portal (Table 1). Patients surviving less than two weeks after surgery were excluded from the survival analysis. Overall survival was assessed as the number of years from surgery to the last follow-up or death. Patients surviving longer than ten years were censored. All patients (including training, validation, and test set) were used for defining the median TLS density thresholds in each tumor type. In cases when several scans were available for one patient, the highest value was selected as the value for that patient.

Patients were defined as TLS-high if their TLS density values were above the tumor type-specific median. The overall survival was analyzed in the context of the clinical and pathological parameters (age, gender, stage) as well as histological parameters (GC or TLS densities) by univariate and multivariate Cox regression (Table 2) (stage and gender were categorized, all other parameters were used as continuous variables), Kaplan–Meier curves and log-rank test. Analyses were done using R version 4.1.2. survival package version 3.4-0.

**Reporting summary.** Further information on research design is available in the Nature Portfolio Reporting Summary linked to this article.

## Results

**Model development.** In this study, we used diagnostic H&E images from three TCGA cohorts, namely LUSC, BLCA, and KIRC, on which we manually annotated TLSs, GCs, and other tissues (commonly termed Rest) (Fig. 1a, b). We used subsets of these annotations for training, validating, and testing the HookNet-TLS model. This model is an improved version of the HookNet model[22], with enhanced computational performance and additional post-processing steps. These steps include generating whole slide segmentation maps (WSI inference), converting segmentation masks into objects, and optimizing object detection thresholds (Fig. 1c).

**Benchmarking TLS and GC manual annotations using immunohistochemistry (IHC).** To assess the quality of H&E-based manual annotations, we used IHC as a reference standard. A trained annotator first annotated an independent set of LUSC and KIRC H&E images from University Hospital Zurich (USZ), which we defined as the blinded annotations. Successively, the same H&E images was re-annotated by making serial sections immunostained for CD20 and CD23 markers available to the annotator, which we defined as the IHC-guided annotations. This aided the objective definition of TLSs as dense CD20+ B cell clusters and centrally located CD23+ follicular dendritic cell networks as a compulsory component of GCs (Fig. 2a). The IHC-guided dataset comprised a total of 639 TLS and 89 GC annotations in the LUSC, and 217 TLS and 57 GC annotations in the KIRC slides. To quantitatively evaluate the performance of the blinded annotation process, we calculated the F1-score as the harmonic mean of precision (positive predictive value) and recall (sensitivity) for the blinded versus IHC-guided objects (see the Methods section for details). We found a high overlap between

**Table 1 Patient characteristics of the analyzed Cancer Genome Atlas (TCGA) cohorts**

| | | | Time to death | | |
|---|---|---|---|---|---|
| Characteristic | N | n (%/range) | HR | 95% CI | p-value |
| KIRC | | | | | |
| Stage | 293 | | | | |
| I+II | | 172 (59%) | – | – | |
| III+IV | | 121 (41%) | 3.62 | [2.34, 5.62] | 0.001 |
| Gender | 293 | | | | |
| FEMALE | | 107 (37%) | – | – | |
| MALE | | 186 (63%) | 1.19 | [0.77,1.84] | 0.43 |
| Age | 293 | 60 (51, 69) | 1.03 | [1.01, 1.05] | 0.004 |
| manual TLS | 293 | 0.43 (0.00, 1.54) | 1.11 | [1.03, 1.19] | 0.006 |
| manual GC | 292 | 0.00 (0.00, 0.00) | 1.27 | [0.64, 2.52] | 0.50 |
| predicted TLS | 293 | 0.60 (0.00, 2.02) | 1.05 | [0.98, 1.13] | 0.15 |
| predicted GC | 293 | 0.00 (0.00, 0.00) | 2.39 | [0.84, 6.80] | 0.10 |

| | | | Time to death | | |
|---|---|---|---|---|---|
| Characteristic | N | n (%/range) | HR | 95% CI | p-value |
| LUSC | | | | | |
| Stage | 332 | | | | |
| I+II | | 269 (81%) | – | – | |
| III+IV | | 63 (19%) | 1.42 | [0.97, 2.06] | 0.069 |
| Gender | 332 | | | | |
| FEMALE | | 88 (27%) | – | – | |
| MALE | | 244 (73%) | 1.15 | [0.81, 1.65] | 0.44 |
| Age | 332 | 68 (63, 74) | 1.01 | [0.99, 1.03] | 0.24 |
| manual TLS | 332 | 5 (2.00, 11.00) | 0.99 | [0.97, 1.01] | 0.25 |
| manual GC | 331 | 0.00 (0.00, 0.41) | 0.94 | [0.83, 1.06] | 0.30 |
| predicted TLS | 332 | 7 (3.00, 13.00) | 0.99 | [0.97, 1.00] | 0.052 |
| predicted GC | 332 | 0.00 (0.00, 0.58) | 0.91 | [0.79, 1.04] | 0.17 |

| | | | Time to death | | |
|---|---|---|---|---|---|
| Characteristic | N | n (%/range) | HR | 95% CI | p-value |
| BLCA | | | | | |
| Stage | 305 | | | | |
| I+II | | 91 (30%) | – | – | |
| III+IV | | 214 (70%) | 3.01 | [1.60, 5.68] | <0.001 |
| Gender | 305 | | | | |
| FEMALE | | 81 (27%) | – | – | |
| MALE | | 224 (73%) | 1.07 | [0.66, 1.71] | 0.79 |
| Age | 285 | 68 (60, 76) | 1.03 | [1.01, 1.05] | 0.007 |
| manual TLS | 305 | 1.7 (0.0, 4.0) | 0.97 | [0.92, 1.04] | 0.41 |
| manual GC | 305 | 0.00 (0.00, 0.61) | 0.80 | [0.63, 1.01] | 0.060 |
| predicted TLS | 305 | 2.4 (0.8, 5.6) | 0.98 | [0.93, 1.04] | 0.48 |
| predicted GC | 250 | 0.00 (0.00, 0.65) | 0.85 | [0.68, 1.06] | 0.15 |

Clinical, pathological and histological parameters of the whole TCGA cohorts (training+validation+test sets) were analyzed in relation with overall survival analyzed by univariate Cox regression. Stage (I+II versus III+IV) and gender (Female vs Male) were categorical and all other parameters were used as continuous variables. Tertiary lymphoid structure (TLS) and germinal center (GC) density was determined as the number of manually annotated or predicted regions per $cm^2$ of the analyzed tissue. Hazard ratio (HR), its 95% confidence interval (CI) and p value for each parameter are reported.

the blinded and IHC-guided TLS annotations (Precision = 0.94; Recall = 0.85; overall F1-score = 0.89), which provided evidence for high accuracy of manual TLS detection in H&E images (Fig. 2b). We found that mainly small TLSs were missed (false negatives) in blinded TLS annotations (Fig. 2c) but these constituted a minor fraction (on average 16%) of IHC-defined TLSs in both tumor types (Fig. 2d). We noticed that in KIRC, a considerable fraction of blinded TLS annotations was classified as false positive (Fig. 2d), mostly consisting in multiple dense lymphocytic clusters which were negative for CD20 staining (Fig. 2e, black arrowheads). To explore the reasons for this inaccuracy, we used multiplex immunofluorescence (mIF) (See Methods section for more details) and found that the tumor microenvironment of KIRC contains frequent T cell aggregates, which are morphologically highly like B cell aggregates in H&E staining. However, due to the lack of a B cell compartment, these are not considered as bona fide TLSs. Based on these findings, we hypothesized that a similar proportion of annotated aggregates in the TCGA KIRC dataset are T cell clusters, and were trained as TLS in the HookNet-TLS model.

Regarding GCs, we found that the accuracy of blinded versus IHC-guided GC annotations was lower than TLS (Precision = 0.95; Recall = 0.40; overall F1-score = 0.57) (Fig. 2b). We found that the main source of the inaccuracy in blinded annotations were false negative rather than false positive detections (Fig. 2d). Image analysis showed that CD23+ cell networks were found in multiple TLS lacking a clear central area of proliferating blasts (Fig. 2a, d, white arrowheads), which is the classical morphological parameter defining GCs in H&E images (Fig. 2a, gray arrowheads). The lack of the classical GC morphology (and thus false negative assessment in the blinded annotations) correlated with smaller GC size (Fig. 2c). These data suggest that GCs in TLSs have a variety of morphologies, which are difficult to detect in H&E images. By extension, we assume that the GC

**Table 2 Prognostic independence of relevant clinical and histological parameters in the analyzed the Cancer Genome Atlas (TCGA) patient cohorts**

|  | HR | 95% CI | *p*-value |
|---|---|---|---|
| *LUSC (manual TLS)* | | | |
| STAGE(III+IV) | 1.448147 | [0.9927, 2.113] | 0.0546 |
| Manual TLS | 0.989205 | [0.9730, 1.006] | 0.1984 |
| *LUSC (predicted TLS)* | | | |
| STAGE(III+IV) | 1.439338 | [0.9876, 2.0977] | 0.0581 |
| Predicted TLS | 0.984524 | [0.9696, 0.9997] | 0.0457 |
| *KIRC (manual TLS)* | | | |
| STAGE(III+IV) | 3.326768 | [2.1327, 5.189] | 1.16e-07 |
| Age | 1.021708 | [1.0025, 1.041] | 0.0267 |
| Manual TLS | 1.057970 | [0.9769, 1.146] | 0.1657 |
| *KIRC (predicted TLS)* | | | |
| STAGE(III+IV) | 3.404224 | [2.1901, 5.291] | 5.22e-08 |
| Age | 1.024197 | [1.0050, 1.044] | 0.0135 |
| Predicted TLS | 1.048095 | [0.9659, 1.137] | 0.2596 |

Clinical, pathological and histological parameters found to have a significant association with overall survival in the whole TCGA cohorts (training + validation + test sets) by univariate Cox regression were tested for their independence by multivariate Cox regression. Stage (I+II versus III+IV) was used as a categorical variable and all other parameters were used as continuous variables. Tertiary lymphoid structure (TLS) and germinal center (GC) density was determined as the number of manually annotated or predicted regions per $cm^2$ of the analyzed tissue. Hazard ratio (HR), its 95% confidence interval (CI) and p value for each parameter are reported.

training set in the TCGA cohort is underrepresented for this diversity.

**Benchmarking HookNet-TLS detection performance**. We compared the output of HookNet-TLS with Faster-RCNN[27], a state-of-the-art approach that has been successfully used in several object detection approaches in a variety of computer vision tasks[33–35]. In contrast to HookNet-TLS, which is a segmentation-based approach, Faster-RCNN predicts bounding boxes around a set of pixels belonging to an object of interest, such as a TLS, which is commonly referred to as a detection-based approach. To assess the performance of Faster-RCNN and HookNet-TLS, we computed the F1-scores within each individual slide for (1) model predictions versus the IHC-guided annotations in the USZ set and (2) model predictions versus the manual annotations in the TCGA independent test set. We found that HookNet-TLS performed significantly better in comparison to Faster-RCNN for detecting TLSs and GCs in both sample sets (Fig. 3a, b). We next computed the overall F1-scores taking all ground truth annotations in each tumor type together. This score does not take into account the true negative slides as there is no means to quantify the absence of a TLS prediction at the annotation level, but this metric is helpful to assess TLS or GC detection in general. The overall F1-score comparison showed that TLSs were best predicted within the LUSC and least within the KIRC samples (Fig. 3c). Comparable results were obtained for GC predictions (Fig. 3d) albeit with reduced performance compared to TLS detection.

Finally, we measured the inter-observer variability on a randomly selected subset of LUSC samples from the TCGA test set (Supplementary Fig. 1). We chose LUSC for this experiment because this cohort showed the highest number of TLS and GC annotations in the test set and thus ensures the highest statistical power for evaluation of both classes. We observed that the F1-scores of HookNet-TLS were in the same range as the F1-scores between two trained annotators (Fig. 3e). The main source of inaccuracy for HookNet-TLS and the reader study were false negative detections, while for Faster-RCNN it was the false positive detections (Supplementary Figure 2a, b). HookNet-TLS showed slightly lower precision (higher false positive predictions) but increased sensitivity for TLS detection in comparison to manual annotators (Supplementary Fig. 2c, d). The overall

F1-scores of the TCGA test set directly correlated with the number of available training annotations for each class and tumor type (Fig. 3f, Fig. 1c) suggesting that further increase of training annotations could improve the predictive performance of HookNet-TLS for both classes. Representative images of HookNet-TLS and Faster-RCNN detection objects are shown in Fig. 3g.

**HookNet-TLS identifies TLS and GC characteristics**. We next investigated different parameters of individual TLSs and GCs, such as size, density and morphology of detected objects by the HookNet-TLS model in different organs. We found that the model detected higher TLS densities in all tumor types than manual annotations (Fig. 4a), in line with the higher rate of false positives (Supplementary Fig. 2b). However, besides prediction errors, there were also true TLS among the false positive predictions, which, via retrospective analysis, we acknowledged as missed in manual annotations (Supplementary Fig. 3). The detected GC density was similar in both approaches (Supplementary Fig. 4a). Overall, TLS density differed significantly across the three tumor types with LUSC showing the highest and KIRC—the lowest densities detected similarly by manual annotations and HookNet-TLS predictions (Fig. 4b). Same results were obtained for the GC densities (Supplementary Fig. 4b). Our previous immunostaining-based quantitative pathology analysis showed that TLSs are larger in tumors with prominent lymphoid neogenesis (TLS-high tumors)[8,10]. HookNet-TLS was superior to manual analysis in detecting such TLS size differences in H&E images (Fig. 4c). This indicates that HookNet-TLS captures the area (shape) of individual TLS with high precision. Finally, in line with previous reports[8,10] we found a significantly higher GC density in TLS-high versus TLS-low tumors by both detection approaches (Supplementary Figure 4c).

By using the Uniform Manifold Approximation and Projection for Dimension Reduction (UMAP)[36], we further explored the main encoding features learned by HookNet-TLS. These features were extracted from the lowest level of the target branch where the field of view of input data is maximized, assuming that those best depict the overall morphological characteristics of each region and its spatial context. We found that TLSs and GCs were encoded similarly across the three different organs and across the different datasets (training, validation and test slides; Fig. 4d), supporting the potential application of HookNet-TLS for pan-cancer TLS analysis. The encoding features were extracted from the central patch of each region, thus the presence of a GC in a TLS may be encoded differently than in a TLS without a GC. To test this, we performed unsupervised PhenoGraph clustering of encoding features for all annotated and predicted TLS and GC regions. We obtained 14 distinct clusters (Fig. 4e), several of which were significantly enriched in the predicted GCs compared to TLSs (Fig. 4f). We next explored the corresponding test set images for TLS predictions assigned to different clusters and confirmed that the majority of TLSs belonging to cluster 7 contained a GC (Fig. 4g), while TLS predictions in cluster 1 occupying the opposite side of the UMAP distribution showed no GCs. Together, these data indicate that HookNet-TLS captures informative morphological features of TLSs and GCs.

**HookNet-TLS predictions bear prognostic relevance**. To evaluate the prognostic relevance of manual TLS annotations and HookNet-TLS predictions we analyzed the corresponding clinical information of the TCGA cohorts (Table 1). Owing to the significant differences in the overall TLS density (the number of TLSs per total tissue area) across the three tumor types (Fig. 4b),

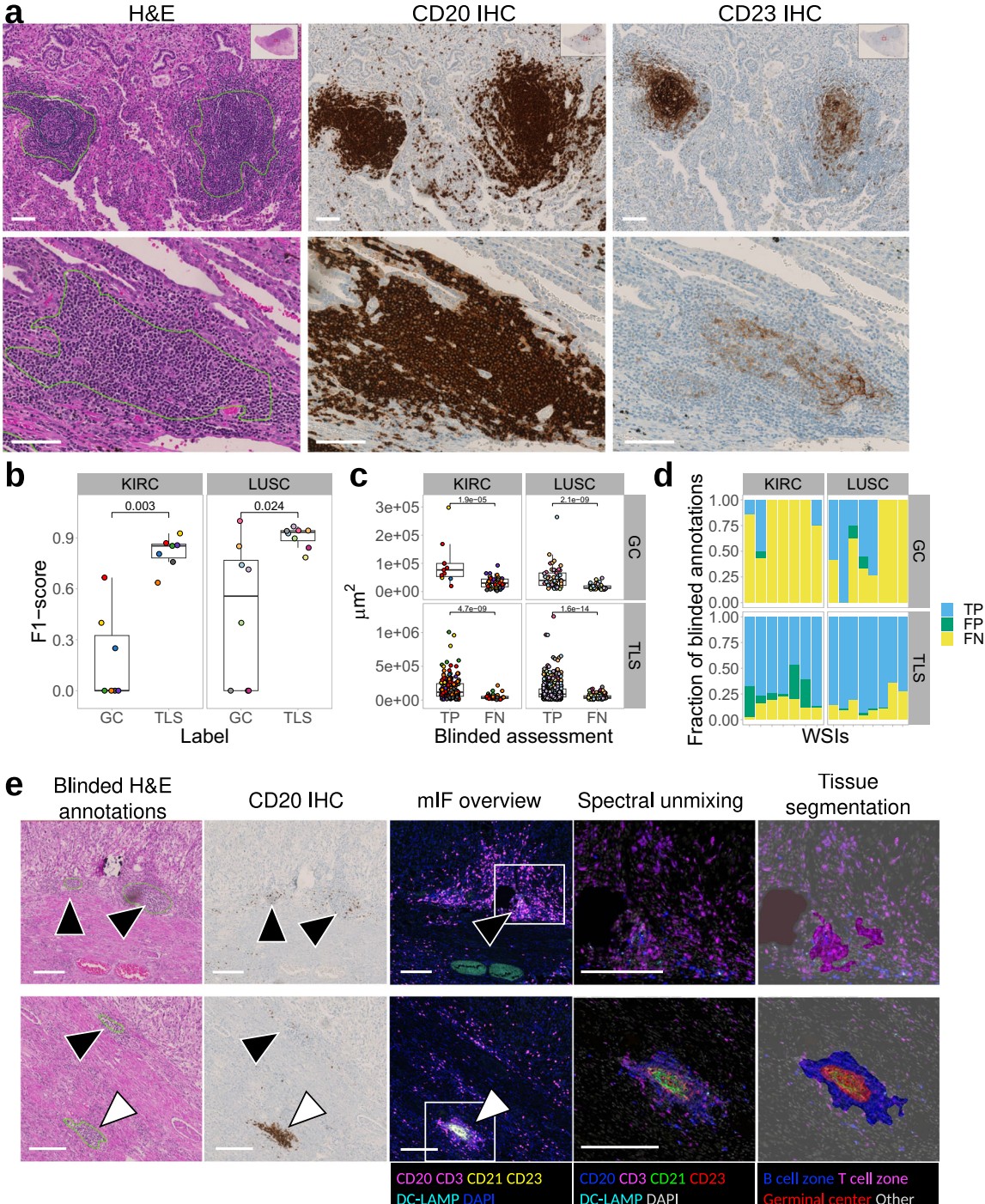

**Fig. 2 Evaluation of manual annotations. a** An independent set of tertiary lymphoid structures TLS-high samples was selected from cohorts of clear cell renal cancer (KIRC) and lung squamous cell carcinoma (LUSC) patients from University Hospital Zurich (USZ). Serial sections were stained by hematoxylin and eosin (H& E) and immunohistochemistry (IHC) for CD20 to detect B cell clusters as a compulsory TLS component, and CD23 to mark follicular dendritic cells within an active germincal centers (GC) reaction. Representative examples of blinded TLS (green lines) and GC (dark blue circle) annotations in H& E images annotated as true positive GCs (gray arrowheads) and false negative GCs (white arrowheads) in accordance with serially stained images for CD20 and CD23 are shown for two LUSC samples. Scale bars = 100 μm. **b** The performance for manual TLS and GC detection was calculated as F1-scores for n=15 independent H& E images (color-coded dots) by assessing the overlap between blinded and IHC-guided annotations. The F1-scores for GC and TLS detection were compared in each tumor type by Mann–Whitney *U* test. **c** The size of IHC-guided annotations that were recognized by blinded analysis as true positives (TP) or false negatives (FN) was compared by Mann–Whitney *U* test. Each dot is an annotation color-coded by the source image. Boxes in (**b**) and (**c**) span across the 25th and 75th percentiles, and whiskers (Tukey style) extend from the hinge to the largest value no further than 1.5 * interquartile range. **d** The proportion of TLS and GC regions categorized as true positives (TP), false positives (FP) or false negatives (FN) by blinded assessment in reference to the IHC-guided annotations in n=15 independent H& E images. **e** The tumor microenvironment composition of KIRC samples was analyzed by multiplex immunofluorescence. Dense B cell and T cell clusters were identified by machine learning algorithm of tissue segmentation of the Inform software (Akoya Biosciences). Examples of false positive TLS (black arrowheads) and false negative GC (white arrowheads) among blinded H& E annotations are displayed in matching regions from H& E, IHC of CD20, and multiplex IF stained sections including zoomed-in regions indicated by the white squares. Scale bars = 200 μm.

we defined survival groups by tumor type-specific median TLS densities based on all slides.

When analyzing the cohort of each tumor type, excluding the training set images, manual TLS density correlated with longer overall survival in LUSC, and with reduced survival in KIRC (Fig. 5a), which is in line with previous results from us and others[8,26]. A trend for a positive survival correlation was also found in BLCA, which reached significance when training samples were included in the analysis (Supplementary Fig. 5a) corresponding to the results from our previous report[10]. The reduced statistical power of the combined test and validation cohort compared to the whole cohort is due to our necessity to include a higher proportion of TLS-high BLCA and KIRC tumors in the training set to achieve a comparable proportion of TLS and GC training annotations among the different tumor types. Previously, it has been shown that the maturation of TLSs towards active GCs was crucial for the beneficial prognosis of TLS density as well as correlated with improved response to immunotherapy in different tumors[8,37–40]. We found that manually annotated TLSs including a GC had no significant survival correlations in LUSC and BLCA, while there were too few patients with GCs in the KIRC cohort for a meaningful comparison. HookNet-TLS predicted total TLS densities showed similar survival correlations in KIRC and had a more prominent survival benefit in LUSC compared to ground truth annotations (Fig. 5a). Furthermore, HookNet-TLS detected GC⁺ TLSs were significantly associated with improved survival in LUSC patients (Fig. 5b). However, HookNet-TLS predictions failed to detect any survival benefit in BLCA either in the test and validation cohort or in the whole cohort (Supplementary Figure 5). To investigate whether this was due to insufficient prediction performance, we implemented a so-called iterative hard-negative mining in the training phase, where false positive predictions on training data were automatically assigned as the Rest class for subsequent training iterations. While a slight increase in the overall F1-scores was achieved, no significant differences were found at the per slide F1-scores (Supplementary Figure 6b) suggesting that the prediction performance cannot be significantly increased without adding further TLS and GC training annotations.

Investigation of HookNet-TLS prediction images did not reveal any differences in quantity or quality of non-TLS region detection (false positive regions) between BLCA and other tumor types (Supplementary Fig. 7), suggesting that the available BLCA whole cohort size is on the lower limit of statistical power to detect the weaker association between BLCA patient survival and TLS density. Finally, we performed multivariate Cox regression analysis for the relevant clinical and histological parameters (Table 1) and demonstrated that predicted but not manual total TLS density was an independent positive prognosticator (as a continuous variable) in LUSC (ground truth annotations $p = 0.2$, predicted TLS $p = 0.046$), while in KIRC, TLS was not independent from the tumor stage or age (Table 2). In line with this, we found that TLS density was increased in the late-stage KIRC samples but not in BLCA or LUSC samples (Supplementary Fig. 8). Together, HookNet-TLS predictions hold similar prognostic capacity as manual TLS assessment in LUSC and KIRC, while an extended BLCA cohort is needed to validate the prognostic correlation of HookNet-TLS predicted regions.

## Discussion

TLSs in the tumor microenvironment are considered novel biomarkers and efforts are underway to standardize their quantification. Our recent analysis suggests that direct TLS quantification in tissue sections through H&E staining or immunostaining is necessary for reliable results[10]. However, recognizing TLS in H&E by visual inspection can be difficult, especially in heavily inflamed tissues yielding a considerable component of subjectivity in manual TLS quantification. To address this issue, we aimed to develop a deep learning model for the detection of TLSs and GCs, to support the quantification of TLS-based biomarkers. Furthermore, the fully automated nature of our approach, and the fact that it solely requires H&E-stained slides, allows objective and reproducible quantification of TLSs on large-scale retrospective and prospective studies based on routinely available data.

When analyzing whole slide H&E images, pathologists use multiple magnifications to identify different cell types and tissue structures. Within the context of TLS quantification, using multiple resolutions and therefore including contextual information is needed to differentiate between regions of high inflammation and TLSs. First, areas with high immune cell density are detected at high resolution. Second, a perimeter where inflammatory cell density drops defining an accumulation rather than continuous infiltration characteristic of unorganized inflammation is assessed at low resolution. HookNet implements such multi-resolution assessment for TLS detection via segmentation. Alternative detection-based methods such as Faster-RCNN predict bounding boxes instead, therefore lacking prediction of morphological features such as area and shape which makes quantification analysis per definition less accurate.

The task of segmenting objects in WSIs is complex due to the vast amount of data and various tissue appearances present. To train the HookNet-TLS model, TCGA slides were exhaustively annotated by three trained researchers, which allowed us to evaluate the model performance on the whole tissue section as well as in comparison to inter-observer variability. Our objective evaluation using IHC as a reference standard demonstrated that manual TLS annotations are, however, imperfect. A fraction of small TLSs was missed contributing to false negative detection of around 16%. Furthermore, false positive TLS detection was caused by T cell clusters present mainly in KIRC, which are indistinguishable from B cell clusters in H&E staining by eye. Whether such T cell clusters represent a precursor stage of TLS is currently unknown. Furthermore, we found that manual GC detection was suboptimal owing to the absence of specific morphologies discernible by H&E staining for smaller GCs. These results revealed the nature of limitations concerning manual TLS detection in H&E images, which also affect the training and the accuracy testing of our model. We observed a direct correlation between the number of available training annotations of TLSs and GCs and the resulting model accuracy when comparing the performance in each tumor type. In an attempt to increase the performance of our model without expanding the training slide set (and thus reducing the independent test set), we developed an iterative training approach with the so-called hard-negative mining, which automatically assigns false positive predictions as the Rest class in subsequent training iterations. While it improved dealing with rare artifacts, it did not achieve a better TLS detection and even reduced the performance for the GC class in the third training iteration (Supplemental Fig. 6). In our IHC-controlled USZ samples as well as from the retrospective examination of TCGA predictions we found that the false positive predictions also contained true TLS missed in manual annotations. Factors such as fatigue when performing large scale analysis, subjective criteria in TLS definitions and lack of IHC data as reference standard contributed to these errors. We showed that HookNet-TLS is capable of encoding morphological differences of each annotation class and that GC-positive TLSs can be identified among the TLS feature clusters. Future work could focus on improving the hard-negative sampling strategy by using

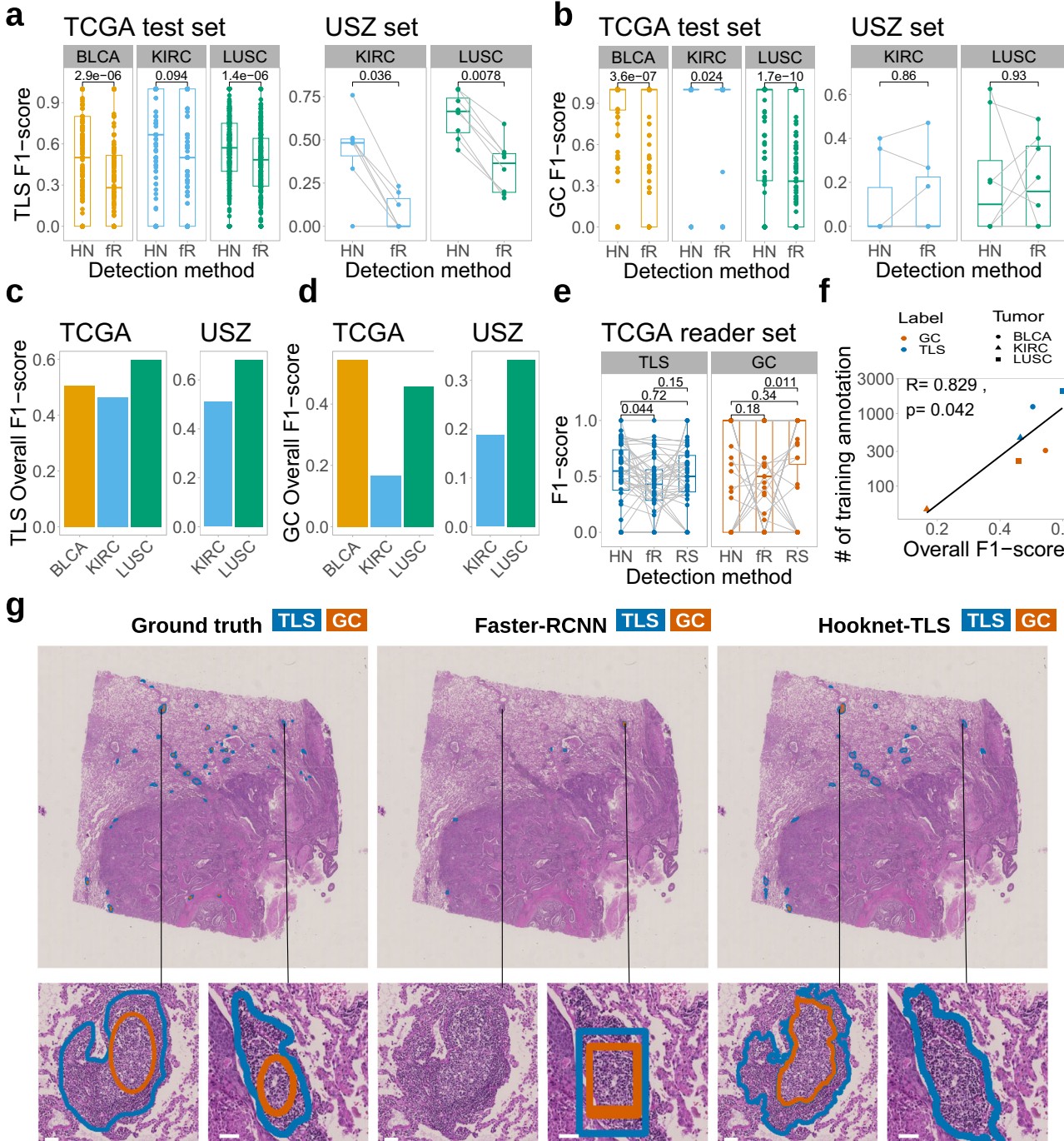

**Fig. 3 Prediction performance of HookNet-TLS model in the Cancer Genome Atlas (TCGA) test set ($n = 767$) and University Hospital Zurich (USZ) set ($n = 15$). a** Per-slide tertiary lymphoid structures (TLS) F1-scores were obtained from the TCGA test set or the USZ set by HookNet-TLS (HN) or Faster-RCNN (fR) models and compared by paired Mann–Whitney $U$ test. **b** Per-slide germinal centers (GC) F1-scores were obtained from the TCGA test set or the USZ set by HookNet-TLS (HN) or Faster-RCNN (fR) models and compared by paired Mann–Whitney $U$ test. **c** Overall TLS F1-scores were obtained from the TCGA test set or the USZ set by HookNet-TLS. **d** Overall GC F1-scores were obtained from the TCGA test set or the USZ set by HookNet-TLS. **e** Per-slide TLS or GC F1-scores were obtained from a random selection ($n = 44$) of slides of the lung squamous cell carcinoma (LUSC) TCGA test set by HookNet-TLS (HN), Faster-RCNN (fR) or inter-observer comparison (the reader study, RS) and compared by paired Mann–Whitney $U$ test. Boxes in (**a**), (**b**) and (**e**) span across the 25th and 75th percentiles, and whiskers (Tukey style) extend from the hinge to the largest value no further than 1.5 * interquartile range. **f**: Spearman correlation analysis was used to determine the association between the overall F1-scores from HookNet-TLS and the number of available training annotations for each class and each tumor type. **g** A representative image from the USZ test set, with corresponding ground truths, detections by Faster-RCNN and HookNet-TLS. Scale bars = 50 μm.

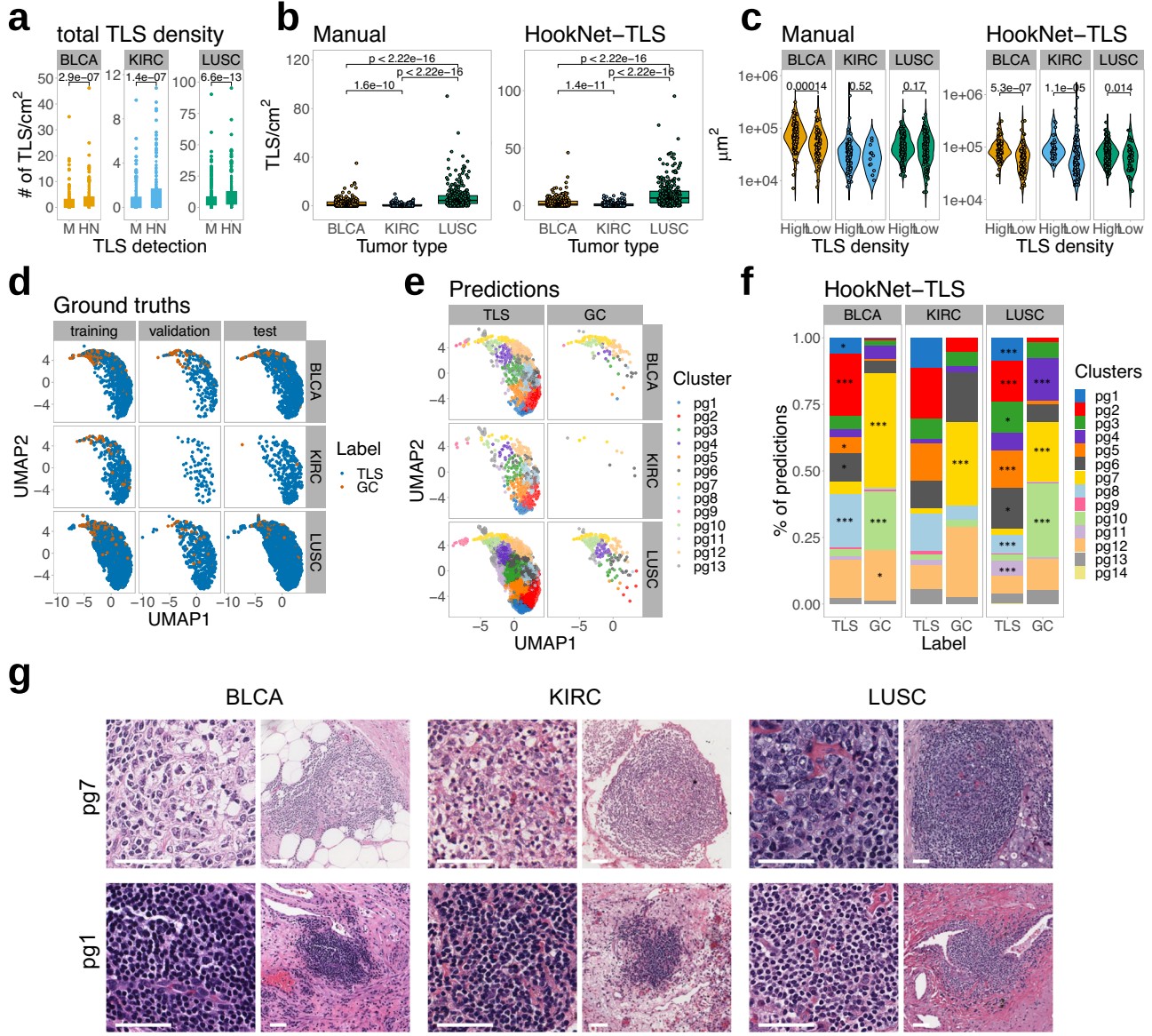

**Fig. 4 Analysis of TLS parameters in different tumors.** TLS parameters were assessed in the Cancer Genome Atlas (TCGA) test set slides (n=767) if not otherwise specified. **a** Pairwise comparison of manually annotated versus HookNet-TLS predicted tertiary lymphoid structures (TLS) density in each image by paired Mann–Whitney $U$ test. **b** TLS density compared across organs by Mann–Whitney $U$ test obtained by manual or HookNet-TLS detection. Boxes in a and b span across the 25th and 75th percentiles, and whiskers (Tukey style) extend from the hinge to the largest value no further than 1.5 * interquartile range. **c** Median TLS density in each tumor type was used as a cutoff to define TLS-high and TLS-low tumors. Average TLS size was compared between the TLS density groups by Mann–Whitney $U$ test. **d** The HookNet-TLS encoding features of all TLS and germinal centers (GC) ground truth annotations and predictions were extracted from all images and subjected to UMAP dimensionality reduction analysis. Data are displayed for ground truth annotations (dots) depicted for each tumor and each image dataset. **e** The HookNet-TLS encoding features of all TLS and GC ground truth annotations and predictions were extracted from all images and subjected to unsupervised clustering using the PhenoGraph algorithm ($k = 50$). Data are displayed for predictions (dots) obtained from the TCGA test set and dots are colored by their cluster assignment. **f** Fractions of HookNet-TLS predictions belonging to different PhenoGraph clusters were compared between TLS and GC classes by Chi square test. Clusters with significant enrichment in either TLS or GC class are marked by asterisks, respectively (Yates corrected $p$-values *<0.05, ***<0.000). Predictions obtained from all images (test+train+validation) were used in this analysis. **g** Representative visualizations of HookNet-TLS predicted TLS in the TCGA test set belonging to PhenoGraph (pg) clusters 1 and 7. A high-resolution central patch and a contextual level are shown for each prediction. Scale bars = 50 μm.

unsupervised clustering of the feature encodings to discern potential true TLSs/GCs from true Rest tissue among the false positive predictions as well as using an expanded TLS and GC training set.

Nevertheless, our model at the initial training iteration showed a similar performance across the different tumors and sample sources and was in the same range as inter-observer variability.

Furthermore, we showed that TLSs are encoded in a similar UMAP space by the model across the different organs, indicating that the model is capable of capturing the TLS features independently of each organ and suggests a potential use-case for pancancer application. Future work implementing IHC-guided annotation process will determine if T cell and B cell clusters may be distinguishable by deep learning, as well as ensure

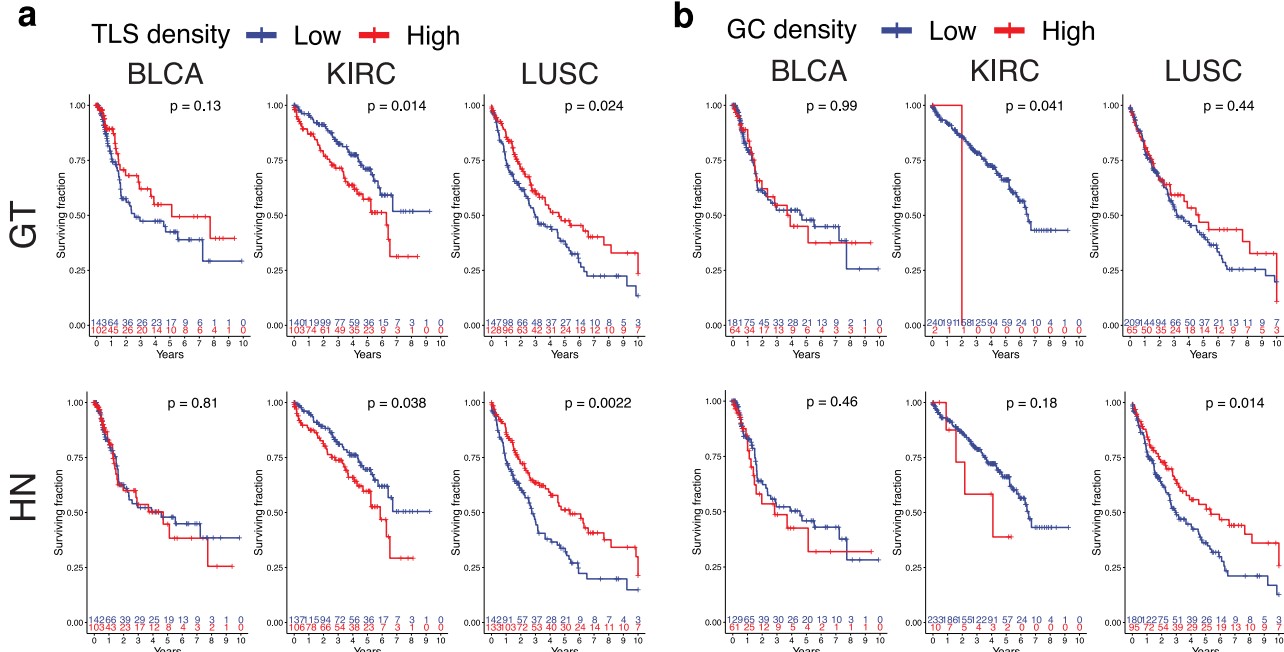

**Fig. 5 Analysis of the prognostic relevance of tertiary lymphoid structures (TLSs) and germinal centers (GCs) in test and validation cohorts (n = 836).** Overall survival was compared between patient groups by Kaplan–Meier curves and log-rank test. **a** TLS-high groups were defined as above median TLS density calculated from whole tumor cohorts of each tumor type for manually annotated (GT) and predicted TLSs (HN). Calculated median manual TLS densities (TLS/cm$^2$) were: clear cell renal cancer (KIRC) 0.437, muscle invasive bladder cancer (BLCA) 1.665, lung squamous cell carcinoma (LUSC) 4.862. Calculated median predicted TLS densities (TLS/cm$^2$) were: KIRC 0.614, BLCA 2.439, LUSC 6.78. **b** Groups were defined by median GC densities of whole cohorts of each tumor type for manually annotated (GT) and predicted TLSs (HN). Due to the low frequency of GCs, this threshold corresponded to GC-negative vs GC-positive tumors in all cohorts. GT, ground truth; HN, HookNet-TLS.

sufficient sampling of TLS and GC variability, which may further improve the robustness and accuracy of the model across different organs.

We performed two manual image analysis (see Methods section) steps before applying the model: (i) exclusion of slides lacking adjacent normal tissue, and (ii) annotation of lymph node areas for exclusion due to their high morphological resemblance with TLS. In the future, these steps may be incorporated into the HookNet-TLS pipeline by using automatic quality control and tissue segmentation[41,42], and a lymph node detector.

Additionally, by incorporating tumor/non-tumor tissue segmentation, TLS density assessment can be further refined in both intra- and peri-tumoral regions. This refinement includes the integration of distance-to-tumor measurements. Such an enriched analysis has the potential to provide more nuanced insights, as the spatial distribution and proximity of TLS to tumor regions may hold additional prognostic information. Furthermore, utilizing this approach allows for the normalization of TLS density based on tumor area or specifically normal area (where most TLS develop), which can offer a more standardized metric that accounts for variations in available tissue size and facilitates more accurate and consistent comparisons across different samples or patient cohorts.

A promising area for enhancement could be researched through the integration with nnUNet, a self-configuring medical image segmentation framework that optimizes U-Net training hyperparameters. The fusion of HookNet-TLS and nnUNet into a single framework could potentially enhance performance by combining their strengths and represents an encouraging topic for future exploration.

We analyzed the HookNet-TLS predicted TLS and GC density in the context of the corresponding clinical data from TCGA. The predicted TLS counts showed similar prognostic associations as

manually annotated TLSs in LUSC and KIRC tumor types, while the prognostic correlations in BLCA were weaker and require a larger sample for reliable assessment. Overall, the correlations between TLS and clinical parameters of the analyzed TCGA cohorts were in line with previously published associations obtained by various different clinical centers: TLS are an independent positive prognosticator in non-small cell lung cancer[8,43–46], a positive prognosticator in BLCA[10,47,48], while untreated KIRC represents a unique exception among various solid tumors, where TLS or B cell infiltration is associated with worse outcome[26,49]. Differences in the extent of TLS maturation have been suggested as a possible explanation for the discrepant prognostic TLS associations in the different tumor types[3]. We also showed significant differences in the total TLS and the mature TLS development across the three organs. This may suggest that parenchymal composition specific to each organ defines the permissiveness of lymphoid neogenesis. In support of this, TLS development was prominent in transplantable melanomas growing intraperitoneally in comparison to the same tumors growing subcutaneously[50]. However, and perhaps more importantly, such differences may also be driven by the intrinsic molecular features of the different tumor types. For example, TLS are more frequent in primary breast tumors expressing HER2 in comparison to HER2-negative ones[51,52], while hormone receptor expression negatively correlates with TLS development[51–54]. Reduced TLS development was found in lung metastasis from renal cancer in comparison to lung metastasis from colorectal cancer, which recapitulates the differences in TLS development at the respective primary sites[55]. Finally, intrinsic tumor antigenicity was also reported as a positive factor for TLS development in colorectal[37,56], bladder[10], and pancreatic cancers[57]. Along the same line, transplantable melanomas overexpressing the model antigen ovalbumin showed higher TLS density than unmodified

tumors[50]. While TLS development can be compared between various conditions and patient groups within each study, the use of different TLS definitions and quantification criteria prevent the comparison of results across different studies. Our developed deep learning model offers objectivity in quantifying TLS in samples from multiple sources and will be useful for result harmonization across studies. We release the source code of HookNet-TLS, as well as the manual annotations of the training set and provide the HookNet-TLS model as an publicly available web-based tool for research use (See Data and Code availability sections for details).

In summary, we presented the HookNet-TLS model, a deep learning approach for detecting TLSs in whole-slide images, which has a superior performance compared to a single-resolution detection approach, and its precision at the current training iteration is within the same range as inter-observer variability. Importantly, our findings revealed that HookNet-TLS predictions hold prognostic relevance and can be used for objective TLS quantification across multiple clinical centers and organs. These qualities are essential for the reliable assessment of TLSs as predictive and/or prognostic biomarkers, which potentially could be implemented in routine pathology workflow in the future.

## Data availability

The TCGA dataset used in this study is publicly available and can be accessed through the TCGA repository https://www.cancer.gov/ccg/research/genome-sequencing/tcga. In addition to the TCGA dataset, we also created annotations that were specifically tailored to our study. These annotations and information on sample inclusion criteria and other study-specific details, can be obtained upon request. Please contact: mart.vanrijthoven@gmail.com to request access to these annotations. Source data for figures can be found in the HookNet-TLS source data[58].

## Code availability

The algorithm and code are open-source available (https://github.com/DIAGNijmegen/pathology-hooknet-tls)[58]. This makes it possible for any researcher to freely experiment with and further develop TLS quantification. Additionally, a live algorithm is accessible on Grand Challenge https://grand-challenge.org/algorithms/hooknet-tls, which allows users to try out the algorithm for free. The model implementation is based on HookNet version 0.0.4 (https://github.com/DIAGNijmegen/pathology-hooknet) and wholeslidedata version 0.0.16 (https://github.com/DIAGNijmegen/pathology-whole-slide-data/). Statistical analyses were done using R version 4.1.2. Rphenograph package version 0.99.1 and ggplot2 package version 3.3.6.

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

## Acknowledgements
This study was funded by the following grants: EXAMODE—EXtreme-scale Analytics via Multimodal Ontology Discovery & Enhancement (nr. 825292), to FC and JvdL, NWO-TTW VIDI project (IGNITE), project number 18388, to FC, Worldwide Cancer Research grant (nr. 18-0629), to MvdB and KS, and Swiss National Science Foundation (SNSF) Prima grant (nr. PR00P3_201656) to KS, SNSF Sinergia grant (CRSII5_177208) to MvdB, SNSF project grant (310030_175565) to MvdB. We thank Dr. Ruben Casanova for his critical advice.

## Author contributions
K.S., F.C. designed the study; M.v.R., K.S., and F.C. wrote the manuscript; S.O., F.P., K.S. generated manual annotations in histology images; M.v.R. designed and programed the HookNet-TLS algorithm, and acquired the computational results; M.v.R. and K.S. analyzed the data; K.S., F.C., J.v.d.L. supervised the study; P.S., H.M., M.v.B. provided access to clinical material; all authors read, gave feedback and agreed to the manuscript.

## Competing interests
J.v.d.L. was a member of the advisory boards of Philips, the Netherlands and ContextVision, Sweden, and received research funding from Philips, the Netherlands, ContextVision, Sweden, and Sectra, Sweden in the last 5 years. He is chief scientific officer (CSO) and shareholder of Aiosyn BV, the Netherlands. F.C. was Chair of the Scientific and Medical Advisory Board of TRIBVN Healthcare, France, and received advisory board fees from TRIBVN Healthcare, France in the last 5 years. He is shareholder of Aiosyn BV, the Netherlands. F.C. is an Editorial Board Member for Communications Medicine, but was not involved in the editorial review or peer review, nor in the decision to publish this article. All other authors declare no competing interests.
