## [Peer Review File · Communications Medicine]

Reviewers' comments:

Reviewer #1 (Remarks to the Author):

In their research article titled "Multi-resolution deep learning characterizes tertiary lymphoid structures in solid tumors," Rijthoven et al. developed a deep learning model called "HookNet-TLS" to quantify Tertiary Lymphoid Structures (TLSs) in cancer. The manuscript is well-written, well-illustrated, and well-structured overall. The authors have included an adequate amount of training and validation data. They have also made their code publicly available and offer an online tool to run their algorithm.

I have some minor comments that I would like to address:

- Several figures may be difficult to read for individuals with red/green color blindness, and this issue should be resolved.
- The design of Figure 3A, B, and E, with the interconnection of data points, makes it challenging to interpret. It would be better if the authors defined the design of the boxplot.
- The prognostic relevance of HookNet-TLS predictions is mentioned "[in a subset of tumors]."
- Regarding Table 1, when closely examining the first entries of the table, the confidence interval is reported as 1.01-1.05, but the p-value is 0.004. Although a Cox regression analysis is appropriate, the specific results of the analysis are not clear to me.
- Please carefully review the manuscript for proper multi-hypothesis testing, as seen in Figure S4C and S8.
- Hugo Symbols should be in italicized capitalized letters.
- The authors seem to calculate TLS densities per area globally. However, it is unclear how they account for differing tumor content in a given Whole Slide Image (WSI).
- Lastly, as a reviewer, I find it challenging to determine whether the title "Multi-resolution deep learning characterizes tertiary lymphoid structures in solid tumors" accurately represents the study's focus. The main emphasis of the paper appears to be the development of a framework for automatically quantifying TLS in solid tumors and associating these quantifications with overall survival.

Reviewer #2 (Remarks to the Author):

The paper titled 'Multi-resolution deep learning characterizes tertiary lymphoid structures in solid tumors' presents an application of the HookNet-TLS model for TLS quantification and identification of germinal centers. The paper is well-written and easy to follow. However, I have some comments:

Firstly, the authors claim to introduce a novel approach by incorporating additional post-processing steps and reducing the computational cost (by decreasing the number of parameters from approximately 50 million to approximately 25 million). The authors need to provide a better explanation of how they achieved this reduction and clarify the specifics of the added post-processing steps. Furthermore, to enable a more comprehensive comparison with other methods, the authors should explain the differences between their developed approach's post-processing steps and those of other models. It is unclear if there are significant novelties compared to the previous HookNet model.

Moreover, it would be beneficial for the authors to compare their model with other models specifically designed and applied to medical applications, beyond just the Faster R-CNN. This would enhance the evaluation and demonstrate the model's effectiveness relative to a wider range of approaches.

Finally, it is essential for the authors to either apply their model to a public dataset or publish the dataset they used for evaluation. This will ensure transparency and reproducibility of the results.

Reviewers' comments:

Reviewer #1

In their research article titled "Multi-resolution deep learning characterizes tertiary lymphoid structures in solid tumors," Rijthoven et al. developed a deep learning model called "HookNet-TLS" to quantify Tertiary Lymphoid Structures (TLSs) in cancer. The manuscript is well-written, well-illustrated, and well-structured overall. The authors have included an adequate amount of training and validation data. They have also made their code publicly available and offer an online tool to run their algorithm.

We would like to extend our sincere thanks for your thoughtful feedback on our manuscript. Your insights have helped us to improve the quality of the paper. To make the changes easily discernible, we've marked all ~~removed text with a red strikethrough~~ and ~~new additions in blue~~, both in this letter and in the revised manuscript.

Q1.1 Several figures may be difficult to read for individuals with red/green color blindness, and this issue should be resolved.

A1.1 In the revised version, we switched from using the official color scheme of the American Association for the Advancement of Science journals to the CUD color scheme¹, which is particularly beneficial for individuals with color blindness.

Q1.2 The design of Figure 3A, B, and E, with the interconnection of data points, makes it challenging to interpret. It would be better if the authors defined the design of the boxplot.

A1.2 We have improved Figure 3 by removing the interconnections in the TCGA dataset boxplots. The used statistical test remains unchanged (paired Mann-Whitney U test). However, we decided to keep the interconnections for the USZ dataset, which has fewer data points. This inclusion offers readers a visual representation to observe the trends between the boxplots.

Q1.3 The prognostic relevance of HookNet-TLS predictions is mentioned "[in a subset of tumors]."

A1.3 We are not fully sure what aspect of our results Reviewer 1 means to address with this statement. In this answer, we consider two possible aspects that this reviewer might be addressing.

Regarding the inclusion of squamous cell lung cancer (LUSC), kidney cancer (KIRC), and bladder cancer (BLCA), a subset of all possible solid tumor types, we included those because these tumors represent entities that have previously been published to have a variety of prognostic correlations for TLS (positive in case of LUSC, negative in case of KIRC, and positive or none in case of BLCA). To improve the clarity, we have supplemented the Introduction section as follows:

... large training datasets as well as independent validation and test sets sourced from The Cancer Genome Atlas (TCGA). ~~As an additional verification, we aimed to assess the prognostic correlations of HookNet-TLS predictions by using~~ ~~We used~~ three tumor cohorts (lung squamous cell carcinoma [LUSC], clear cell renal cell carcinoma [KIRC], and muscle invasive bladder cancer [BLCA]) with known ~~differences in their~~ prognostic associations for TLSs [8, 10, 26]. ~~We and~~ generated manual annotations of TLSs and GCs based on visual interpretation of H&E staining ~~and~~ ~~We~~ benchmarked such annotations by matched IHC...

Regarding validating the prognostic value of HookNet-TLS on subsets of included cohorts, namely the unseen test sets, this was done to have the most objective evaluation using samples that HookNet-TLS was never exposed to during training. This is a common procedure in machine learning and statistics, and we consider it as

¹ <http://people.apache.org/~crossley/cud/cud.html>

a fair evaluation of the performance of our method. At the same time, to provide a complete overview of the potential of TLS as a biomarker, we also reported its prognostic value assessed on the full cohort, also including cases that were used to train the segmentation model.

For the sake of clarity, we have further addressed this point in the paper by adding the following sentence to the Method section:

TCGA data

... divided into three independent datasets for model training (n=188), validation (n=69), and testing (n=767), respectively (Figure 1). This allowed for an objective analysis of test cases that were not exposed during training.

Q1.4 Regarding Table 1, when closely examining the first entries of the table, the confidence interval is reported as 1.01-1.05, but the p-value is 0.004. Although a Cox regression analysis is appropriate, the specific results of the analysis are not clear to me.

A1.4 We understand that this reviewer has concerns regarding the small confidence interval of the reported hazard ratio (HR). This is associated with the fact that the analyzed parameter is used as a continuous variable instead of a categorical variable (for example, two groups with high versus low values), which is mentioned in the method section. The HR increment for each continuous variable is much smaller than when two extreme groups are compared. The significant p-value for the continuous variable indicates the consistency of HR association with the changing values of the continuous variable.

Q1.5 Please carefully review the manuscript for proper multi-hypothesis testing, as seen in Figure S4C and S8.

A1.5 Thank you for noticing this potential issue. We have reviewed the analyses and concluded that Figure S4C does not contain multiple tests as each comparison is made on independent samples (three tumor types) with independent measurements (manual annotation and HookNet-TLS predictions), which define their own cohort medians (thus the High and Low groups in each measurement contains not identical sample sets). Figures S6 and S8 would require a partial adjustment for multiple testing as the “middle” groups were compared twice, although against a different group each time. We have added the clarification in the figure legends that no correction for multiple testing was performed for these comparisons, as follows:

S6: ... P values were obtained by paired Mann-Whitney U test. No correction for multiple testing was applied.

S8: ... No correction for multiple testing was applied.

Q1.6 Hugo Symbols should be in italicized capitalized letters.

A1.6 We have carefully checked this, and based on the HUGO Gene Nomenclature Committee², we found out that the italicized Hugo Symbols refer specifically to genes, alleles, and RNAs to distinguish them from proteins. Since in this manuscript, we have referred to only proteins, we concluded that the HUGO symbols should not be italicized.

Q1.7 The authors seem to calculate TLS densities per area globally. However, it is unclear how they account for differing tumor content in a given Whole Slide Image (WSI).

A1.7 We appreciate this feedback regarding the calculation of TLS densities per area and the consideration of differing tumor content in Whole Slide Images. We acknowledge that these aspects need further attention. In our

² <https://www.ncbi.nlm.nih.gov/pmc/articles/PMC7494048/>

study, we focused on the global quantification of TLS densities per area because there is little to no standardization in grossing and cutting in clinical practice when preparing tissue for diagnosis - excised tumors are cut into chunks, embedded, and analyzed by histology, and thus the ratio of tumor and adjacent normal tissues in such blocks is random across a cohort of patients. In this study, we specifically selected diagnostic images, which contained tumor and adjacent normal tissue, to ensure analysis of the region with the highest probability to contain TLS (as TLSs develop in the tumor periphery for the selected tumor types). But we recognize the need for future work to address the potential effects of the available area of tumor and normal tissue for the assessment of TLS density. To address this aspect, we have extended the discussion section on this topic, where we outlined the necessity of developing methodologies to account for the variability in normal/tumor content when quantifying TLS densities, which we see as the next step in our future research. Here below is the modified text.

~~“Additionally, tumor/non-tumor tissue segmentation may be incorporated for refined TLS density assessment in intra- and peri-tumoral regions including distance-to-tumor measurements, which may hold additional prognostic and/or predictive information”~~

Additionally, by incorporating tumor/non-tumor tissue segmentation, TLS density assessment can be further refined in both intra- and peri-tumoral regions. This refinement includes the integration of distance-to-tumor measurements. Such an enriched analysis has the potential to provide more nuanced insights, as the spatial distribution and proximity of TLS to tumor regions may hold additional prognostic information. Furthermore, utilizing this approach allows for the normalization of TLS density based on tumor area or specifically normal area (where most TLS develop), which can offer a more standardized metric that accounts for variations in available tissue size and facilitates more accurate and consistent comparisons across different samples or patient cohorts.

Q1.8 Lastly, as a reviewer, I find it challenging to determine whether the title "Multi-resolution deep learning characterizes tertiary lymphoid structures in solid tumors" accurately represents the study's focus. The main emphasis of the paper appears to be the development of a framework for automatically quantifying TLS in solid tumors and associating these quantifications with overall survival.

A1.8 We appreciate your comment. We would like to defend our choice of the title as HookNet-TLS is not the only automated approach for TLS quantification. In this work, we have developed a unique algorithm that is based on multi-resolution analysis, which we showed to achieve superior detection performance compared to traditional single-resolution approaches. The prognostic correlation is used as a final means to determine the performance of the model in detecting relevant histological features but is not the only TLS parameter that we analyzed. Nevertheless, since the ultimate intended use of the model would be to determine TLS correlations with patient survival or response to therapy, we adjusted the title following your suggestion

“Multi-resolution deep learning characterizes tertiary lymphoid structures **and their prognostic relevance** in solid tumors”.

Reviewer #2

The paper titled 'Multi-resolution deep learning characterizes tertiary lymphoid structures in solid tumors' presents an application of the HookNet-TLS model for TLS quantification and identification of germinal centers. The paper is well-written and easy to follow. However, I have some comments.

Thank you for your valuable comments and suggestions. We appreciate the insights provided, which have assisted us in improving the clarity and depth of our manuscript. To make the changes easily discernible, we have marked all ~~removed text with a red strikethrough~~ and **new additions in blue**, both in this letter and in the revised manuscript.

Q2.1 Firstly, the authors claim to introduce a novel approach by incorporating additional post-processing steps and reducing the computational cost (by decreasing the number of parameters from approximately 50 million to approximately 25 million). The authors need to provide a better explanation of how they achieved this reduction and clarify the specifics of the added post-processing steps. [...] Furthermore, to enable a more comprehensive comparison with other methods, the authors should explain the differences between their developed approach's post-processing steps and those of other models. It is unclear if there are significant novelties compared to the previous HookNet model.

A2.1 We have addressed both of these comments by elaborating on how the reduction in computational cost was achieved, clarifying the specifics of the additional post-processing steps, and explaining the differences between the post-processing steps of our developed approach and those of other models, including the original HookNet model.

Below we show the updated text:

...
The reduction was made by reducing the number of filters in each layer of the neural network, which empirically did not have a significant impact on the performance.
...

~~Moreover, unlike cite{van_rijthoven_hooknet_2021}, we implemented the generation of WSI segmentation and confidence maps as well as extraction of detected objects from these maps. We used a sliding window approach, which involved extracting all tiles covering the WSI, predicting the segmentation output for each tile, and stitching all the tiles together into the WSI map, whereafter we used the `wholeslidedata` package cite{van_rijthoven_wholeslidedata_2023} to extract TLS and GC detection objects via contour finding with the `opencv2` python package.~~

In contrast to the approach taken in the original HookNet model, our implementation encompasses three major extensions: 1) generation of Whole Slide Image (WSI) segmentation maps, 2) generation of confidence maps, and 3) extraction of detected objects from these maps. Specifically, we employ a sliding window approach, wherein tiles spanning the entire WSI are extracted and subjected to segmentation predictions. Subsequently, these tiles are assembled to construct the complete WSI segmentation map. For the extraction of TLS and GC detection objects, we utilize the 'wholeslidedata' package in conjunction with contour finding through the 'opencv-python' package. Through these extensions, we effectively transform the original HookNet segmentation model into a detection model.

~~Object filtering~~ Post-processing

....
Faster-RCNN produces a single probability per object. We tuned object thresholds by optimizing the F1 score on the validation set. Standard Faster-RCNN uses non-maximum suppression to reduce overlapping detections, and overlapping objects resulting from the sliding window method, used for full WSI detection, were merged.
Finally, because GC objects are predicted independently from TLS objects, we eliminated GC-predicted objects that did not overlap with at least 50% of a TLS.

Q2.2 Moreover, it would be beneficial for the authors to compare their model with other models specifically designed and applied to medical applications, beyond just the Faster R-CNN. This would enhance the evaluation and demonstrate the model's effectiveness relative to a wider range of approaches.

A2.2 This reviewer's concern regards the comparison between the proposed approach and other models commonly applied in medical imaging applications. We would like to highlight that in our previous research: "HookNet: Multi-resolution convolutional neural networks for semantic segmentation in histopathology whole-slide images"³, we have extensively compared the HookNet model with U-Net, which is a widely acknowledged model in medical image segmentation. We have tailored the U-Net model into HookNet specifically for histopathology images. In this current study, the comparison with Faster R-CNN aims to demonstrate the advantages of detection-via-segmentation over direct detection approaches. In light of your suggestion, we have explicitly mentioned our comparison with U-Net in the paper. Below, we show the added text in the introduction and mention the comparison against U-Net:

Our recently developed AI model called HookNet successfully segmented TLSs and GCs as well as other tissue structures in H&E-stained WSIs of lung cancer. HookNet has been previously extensively compared to U-Net, a widely acknowledged segmentation model for medical image analysis, and demonstrated superior performance. The unique feature of HookNet is its integration of multiple image resolutions to produce segmentation output.

Based on your comment, we also have started considering the utilization of nnUNet, a highly successful framework for medical image analysis that leverages a self-configuring approach to optimize U-Net training hyperparameters. While developing a "nnHookNet" model is beyond the scope of this paper, we acknowledge the potential of exploring this approach in future work. Therefore, we have introduced a dedicated section in the discussion to highlight this idea, aiming to inspire and motivate future research to investigate the application of nnUNet in conjunction with HookNet-TLS. We have added the following text to the discussion:

A promising area for enhancement could be researched through the integration with nnUNet, a self-configuring medical image segmentation framework that optimizes U-Net training hyperparameters. The fusion of HookNet-TLS and nnUNet into a "nnHookNet-TLS" framework could potentially enhance performance by combining their strengths and represents an encouraging topic for future exploration.

Q2.3 Finally, it is essential for the authors to either apply their model to a public dataset or publish the dataset they used for evaluation. This will ensure transparency and reproducibility of the results.

A2.3 Thank you for your comment regarding the accessibility of the dataset used for evaluation. We recognize the importance of transparency and reproducibility in research. We would like to clarify that the TCGA data used in our study is already publicly available and that this is acknowledged in the Data Availability statement of our paper. As for the annotations, we are committed to sharing them, and they will be available upon request to facilitate the research endeavors of fellow researchers; this is also acknowledged in the Data Availability section of the paper.

³ <https://www.sciencedirect.com/science/article/pii/S1361841520302541>

REVIEWERS' COMMENTS:

Reviewer #1 (Remarks to the Author):

Thank you for addressing my comments, and congratulations on your manuscript.

Reviewer #2 (Remarks to the Author):

I would like to express my gratitude to the authors for providing their responses. However, I must admit that I am unsure if this paper can be considered as a work that proposes a novel framework for the automated quantification of TLS in solid tumors. One of the main reasons for my uncertainty is the lack of an extensive comparison of their method with other existing approaches. My suggestion for the authors would be to define a clearer scope of what they intend to present in their work.

Reviewers' comments:

Reviewer #1 (Remarks to the Author):

Thank you for addressing my comments, and congratulations on your manuscript.

We sincerely appreciate your positive feedback and thank you for your constructive comments which guided our revisions.

Reviewer #2 (Remarks to the Author):

I would like to express my gratitude to the authors for providing their responses. However, I must admit that I am unsure if this paper can be considered as a work that proposes a novel framework for the automated quantification of TLS in solid tumors. One of the main reasons for my uncertainty is the lack of an extensive comparison of their method with other existing approaches. My suggestion for the authors would be to define a clearer scope of what they intend to present in their work.

We value your feedback and the perspective you provided regarding the novelty and comparison of our method. Upon reflection, we agree with your assessment. In the revised manuscript, we have toned down claims of novelty and removed any mentions of introducing a "novel framework". We've concentrated on detailing the utility and findings of our method for the quantification of TLS in solid tumors. We hope that these adjustments provide a clearer scope of our work and address your concerns.